# Integrative Taxonomy of the Spinous Assassin Bug Genus *Sclomina* (Heteroptera: Reduviidae: Harpactorinae) Reveals Three Cryptic Species Based on DNA Barcoding and Morphological Evidence

**DOI:** 10.3390/insects12030251

**Published:** 2021-03-16

**Authors:** Ping Zhao, Zhenyong Du, Qian Zhao, Donghai Li, Xiaolan Shao, Hu Li, Wanzhi Cai

**Affiliations:** 1Key Laboratory of Environment Change and Resources Use in Beibu Gulf (Ministry of Education) and Guangxi Key Laboratory of Earth Surface Processes and Intelligent Simulation, Nanning Normal University, Nanning 530001, China; zpyayjl@126.com; 2MOA Key Lab of Pest Monitoring and Green Management, Department of Entomology, College of Plant Protection, China Agricultural University, Beijing 100193, China; caudzy@126.com (Z.D.); zhaoq919@163.com (Q.Z.); 3Department of Plant Protection, Kaili University, Kaili 556000, China; ldh19970406@163.com (D.L.); Shaoxiaolan8@163.com (X.S.)

**Keywords:** *Sclomina*, DNA barcoding, species delimitation, phylogeny, cryptic species, biology, Rubus

## Abstract

**Simple Summary:**

The assassin bugs (Heteroptera: Reduviidae) are one of the largest and morphologically most diverse families of true bug, having essential impacts on forest ecosystems as predators. The spinous reduviid genus *Sclomina* exhibits shape mimicry and protective coloration adapted to the spinous *Rubus* plant that they inhabit. The genus *Sclomina* shows gradual morphological variability, so its morphological classification is still unresolved, and its biology is almost unknown. In this study, DNA barcodes and morphological evidence were combined to accurately divide the species of a comprehensive collection sampled in South China and North Vietnam. We found three cryptic species. The biological information and mimicry behavior uncover their successive evolutionary survival strategies.

**Abstract:**

*Sclomina* Stål, 1861 (Heteroptera: Reduviidae: Harpactorinae) is endemic to China and Vietnam, with only two species, *Sclomina erinacea* Stål, 1861 and *Sclomina guangxiensis* Ren, 2001, characterized by spinous body and dentate abdominal connexivum. However, due to variable morphological characteristics, *Sclomina erinacea*, which is widely distributed in South China, is possibly a complex of cryptic species, and *Sclomina guangxiensis* was suspected to be an extreme group of the *S. erinacea* cline. In the present study, we conducted species delimitation and phylogenetic analyses based on the mitochondrial cytochrome c oxidase subunit I (*COI*) gene sequences of 307 *Sclomina* specimens collected from 30 sampling localities combined with morphological evidence. The result showed that all samples used in this study were identified as five species: *Sclomina guangxiensis* is a valid species, and *Sclomina erinacea* actually includes three cryptic species: *Sclomina xingrensis* P. Zhao and Cai sp. nov., *Sclomina pallens* P. Zhao and Cai sp. nov., and *Sclomina parva* P. Zhao and Cai sp. nov. In this paper, the genus *Sclomina* is systematically revised, and the morphological characteristics of the five species are compared, described, and photographed in detail. We elucidate the evolutionary history of *Sclomina* based on results of estimated divergence time. The body shape and coloration (green in nymph and brown in adult) of *Sclomina* match their environment and mimic the *Rubus* plants on which they live. The symbiotic relationship between *Sclomina* and spinous *Rubus* plants is presented and discussed.

## 1. Introduction

Assassin bugs are a widely morphologically diverse group and important enemy insect in ecosystems [1]. The members in *Sclomina* considered in this study not only possess unique phenotypes in body coloration and shape, mimicking *Rubus* plants, they are the most common insect predators in forests in South China. *Sclomina* Stål, 1861 is a small genus of the subfamily Harpactorinae (Heteroptera: Reduviidae) with the well-known *Sclomina erinacea* Stål, 1861 as type species, which was set up based on a specimen collected from Hongkong, China [2,3,4,5]. Ren found the second species *Sclomina guangxiensis* Ren, 2001 only narrowly distributed in the southwestern region of Guangxi province in China [6]. Up to the present study, two species were known in this genus and endemic to South China and Northern Vietnam. During the examination of many *Sclomina* specimens kept in China Agricultural University and from field collection in recent years, we found that the body structure and coloration of the species *Sclomina erinacea* in the different sampling localities are distinctly different, especially the structure of the abdominal connexivum, the spines on the head and pronotum, and the spine-shaped processes on the endosoma of male genitalia. The phenotypic differences are distinct in different geographical populations of *Sclomina erinacea*. The second species, *Sclomina guangxiensis,* with narrow distribution, is unique in morphological structure, but little is known about whether it is an extreme case of a *Sclomina erinacea* cline. Performing identification based on complicated morphological characteristics is difficult; therefore, we used DNA barcoding to analyze the genetic distance and phylogenetic relationships among different samples to improve species identification. 

DNA barcoding is a system that uses short, standard gene sequences as inter-species labels to perform rapid, accurate, and automatic species identification [7,8,9]. The mitochondrial cytochrome *c* oxidase subunit I (*COI*) gene sequence is a powerful marker for the discrimination of evolutionary divergence because of its female inheritance, high evolutionary rate, and stable gene content [10,11]. *COI* has been proven to be an appropriate DNA barcoding marker in the animal kingdom due to its relatively easy amplification and fewer numbers of insertions and deletions [12], also in suborder Heteroptera [13,14,15]. Many studies based on *COI* DNA barcoding sequences have shown its strength in species identification and defining the cryptic species in known species [16,17,18,19,20,21,22].

To accurately determine the taxonomy of the genus *Sclomina*, we sequenced the mitochondrial DNA barcoding sequences of 307 individuals from 29 different geographical populations of *Sclomina erinacea* and one population of *Sclomina guangxiensis* to analyze the genetic divergence and reconstruct the phylogenetic relationships to provide molecular evidence for species delimitation. Based on both molecular and morphological evidence, we tried to confirm whether *Sclomina guangxiensis* is a valid species, and whether there are cryptic species in *Sclomina erinacea*. We combined the estimated divergence time, phylogeny, and biogeography to reconstruct the evolutionary history of *Sclomina*. We also conducted a biological study and recorded the field plant on which the *Sclomina* bugs live, then discussed the symbiotic relationship between *Sclomina* and Rubus plants.

## 2. Materials and Methods

### 2.1. Specimens and Acronyms

This study was based on the examination of over 500 specimens, including the specimens of *Sclomina erinacea* Stål, 1861 kept in the Natural History Museum (BMNH); the type specimens of *Sclomina guangxiensis* Ren, 2001 (NKU); the other specimens of the genus *Sclomina* Stål, 1861 kept in China Agricultural University (CAU); and the newly collected material deposited in Nanning Normal University (NNU) and Kaili University (KU). The following acronyms for public collections are used:BMNH, Department of Entomology, Natural History Museum, London, U.K.;CAU, Entomological Museum of China Agricultural University, Beijing, China;NKU, College of Life Sciences, Nankai University, Tianjin, China;NNU, Nanning Normal University, Nanning, Guangxi, China;KU, Department of Plant Protection, Kaili University, Kaili, Guizhou, China.

Molecular specimens were collected from 30 sampling localities in Southern China and Vietnam (Table A1). All specimens were preserved in 100% ethanol and stored at −20 ℃ until DNA extraction. More than 360 specimens were used for the molecular study, and we finally obtained the mitochondrial *COI* sequences (658 bp) of 307 individuals (Alignment S1). All sample codes used in present study are shown in Table A1. The GenBank Accession numbers for every sample’s individual code are shown in Appendix A.

### 2.2. Taxonomy

The external structures were examined using a binocular dissecting microscope. Male genitalia were soaked in hot 90% lactic acid for approximately 10 min to remove soft tissue, then in hot distilled water, and dissected under a microscope. The abdomen was soaked in hot 5% NaOH for approximately 10 min to remove soft tissue, then cleaned in hot distilled water, and dissected under the microscope. The clean abdominal body wall was placed on the slide, pressed flat by the cover slide, and photographed using an Olympus SZX7 trinocular microscope (Olympus Corporation, Tokyo, Japan) and Canon D60 SLR camera (Canon Inc., Tokyo, Japan). The dissected parts of the genitalia structures and abdomen were placed in a plastic microvial with lactic acid for the corresponding specimen. All photos of male genitalia were taken with the aid of Research Stereo Microscope SMZ25 (Nikon Corporation, Tokyo, Japan). Measurements were obtained using a calibrated micrometer. Body length was measured from the apex of the head to the tip of the hemelytra in the resting position. The maximum width of the pronotum was measured across humeral angles. All measurements are provided in millimeters. The classification system and morphological terminology mainly follow those of Hsiao and Ren (1981) and Lent and Wygodzinsky (1979) [4,23]. The species names in the text are arranged alphabetically.

### 2.3. Integrative Molecular Analysis Based on DNA Barcodes

#### 2.3.1. DNA Extraction and Sequencing

We used thorax muscles of a single adult or the whole body of a single nymph for DNA extraction. The total DNA was extracted using the Animal Tissue Extraction Kit (Golden EasyPure^®^ Genomic DNA Kit, Transgenbiotech, Beijing, China) and stored at –20 ℃ in a refrigerator. From newly collected material, the *COI* gene was amplified using the developed primers *COI*-F (5′-GCCAGACATAGCATTTCCCC-3′) and *COI*-R (5′-ATTGCAAATATAGCTCCCATGGA-3′). The PCR reaction system was 50 μL: 1 μL DNA template, 25 μL 2× EasyTaq^®^PCR SuperMix (Transgenbiotech, Beijing, China), 1 μL forward and reverse primers (10 μM), and 22 μL nuclease-free water. The reaction procedure was as follows: pre-denaturation at 94 ℃ for 3.5 min, denaturation at 94 ℃ for 30 s, annealing at 55 ℃ for 30 s, elongation at 72 ℃ for 1 min, and 35 cycles and elongation at 72 ℃ for 8 min. The PCR products were detected by 2% agarose gel electrophoresis, then observed and photographed in the gel imaging analysis system. For all specimens, sequences were obtained and bidirectionally sequenced using the same PCR primer pairs. Guangzhou Qingke Biotechnology Co., Ltd. (Guangzhou, China) and Shanghai Shenggong Biotechnology Co., Ltd. (Shanghai, China) completed the sequencing of PCR amplicons using the Sanger sequencing method.

#### 2.3.2. Phylogenetic Analyses

A total of 307 *COI* sequences were successfully obtained from over 360 samples of *Sclomina* (Alignment S1). In addition, four outgroup *COI* sequences were downloaded from GenBank and used in the phylogenetic analyses (Table A1). Raw forward and reserve sequences were assembled and edited using BioEdit version 7.0.9.3 [24]. The resultant sequences were 658 bp and were identified as the *COI* sequences of the genus *Sclomina* in the BLAST procedure search in GenBank. The DNA sequences were aligned using MEGA X [25]. For all aligned *COI* sequences, a neighbor-joining (NJ) tree was used to estimate the topologies using MEGA X based on the Kimura-2-Parameter (K2P) model [26] and 1000 bootstrap replicates. The monophyletic for the delimited species was defined by NJ analyses. 

The maximum likelihood (ML) tree was constructed using the IQ-TREE web server [27] with 1000 replicates using the ultrafast bootstrap approximation approach. The substitution model was automatically predicted in the IQ-TREE software and we used the best-fit model of TIM2 + F + I + G4 according to the Bayesian information criterion (BIC). 

The Bayesian inference (BI) was conducted using MrBayes 3.2.2 [28] with two simultaneous Markov chain Monte Carlo (MCMC) runs of 2 million generations and tree sampling every 1000 generations. The GTR + I + G model was used in the BI analysis based on the prediction of PartitionFinder 2 [29] according to BIC. The first 25% of trees were discarded as burn-in.

#### 2.3.3. Species Delimitation

We used two different species-delimitation methods for the *COI* dataset of *Sclomina.* In the DNA barcoding gap analysis, according to the result of the five delimited species from 307 individuals defined by the phylogenetic topology of the NJ, ML, and BI trees; morphological characteristics; and geographical distribution information, we used the Kimura-2 parameter (K2P) model [26] to compute intraspecific and interspecific pairwise genetic distances among the 307 *COI* sequences of the five species in MEGA X (Appendix A). 

The automatic barcoding gap discovery (ABGD) analysis does not require the prior partition of species. In ABGD analysis, the molecular operational taxonomic units (MOTUs) partitioned from the *COI* dataset of *Sclomina* were delimited based on pairwise genetic distances by ABGD software [30]. The analysis was conducted based on the Kimura (K80) TS/TV 2.0 model. The steps value was 20. For all other parameter values, we used the default: X (relative gap width) was 1.5; prior intraspecific divergence was 0.001–0.1 (Pmin–Pmax); the Nb bins (for distance distribution) value was 20.

#### 2.3.4. Estimation of Divergence Time

The divergence time was estimated using BEAST 2.6.1 software from the CIPRES online website [31,32]. The XML files running in BEAST 2 were configured in BEAUti. Due to the lack of fossil records, we used molecular clock calibration and the 1.77% nucleotide substitution rate of the *COI* gene per million years [33]. The GTR + I + G substitution model, birth–death prior model, and uncorrelated exponential relaxed clock model were used in the estimation of divergence time. Two independent MCMC chains were run for 10 million generations and stored per 10,000 generations. Tracer 1.7 [34] was used to check the effective sample size (ESS), which was larger than 200 for all statistic parameters. The maximum credibility tree was generated using the TreeAnnotator program with 25% of trees as burn-in.

### 2.4. Biological Study

The 7 female and 5 male alive specimens (sample code: GZLS, Table A1) of *Sclomina erinacea* collected from Fangxiang country, Leishan town, Guizhou province, China, were reared under laboratory conditions (26 ± 2 ℃, natural light) and fed on fruit flies. The annual life history was recorded. The egg and 1–5-instars nymphs were described and are illustrated.

The spinous assassin bug genus *Sclomina* prefers to inhabit plants with spines and glandular hairs, especially *Rubus*. The symbiotic plant species were recorded in 14 collecting localities (Table A1). Plant species identification was performed using plant recognition software (http://www.iplant.cn/hbl, accessed on 15 March 2021) and a website database (http://www.iplant.cn, accessed on 15 March 2021).

## 3. Results

### 3.1. Phylogenetic Relationships

In total, 307 individuals were included in the *COI* dataset, 852 bp sequences were firstly aligned, after trimming both ends of the alignment, 658 bp of the standard DNA barcoding sequences were used for the analyses. In the sequence matrix, the number of variable sites was 135 with 124 parsimony-informative sites. Phylogenetic trees constructed using three approaches (BI, ML, and NJ) inferred from *COI* dataset showed no major differences and highly supported the monophyly of *Sclomina* Stål, 1861 (Figure 1). 

The highly supported clade of *Sclomina guangxiensis* Ren, 2001 was clearly distinct from other species of *Sclomina* with a high divergence level. *Sclomina guangxiensis* is a valid species and the basal part of ingroups in the phylogenetic tree (Figure 1a,b, clade E, Figure 2p and Figure 3p). The remaining individuals formed two clades, which correspond to two morphological groups: spine-shaped connexivum (Figure 1a,b, clade A, Figure 2a–o and Figure 3a–o) and Y-shaped connexivum (Figure 1a,b, clades B–D, Figure 2q–t and Figure 3q–t). In comparison to the three closely related cryptic species clustering from five locality populations within the Y-shaped connexivum group (Figure 1a,b, clades B–D), *Sclomina xingrensis* sp. nov. (Figure 1a,b, clade B), *Sclomina parva* sp. nov. (Figure 1a,b, clade C), and *Sclomina pallens* sp. nov. (Figure 1 a,b, clade D), the divergence level between three new species was much higher (Figure 1; Appendix A). The 6 GZAL and 13 GXTL samples (for sample codes, see Table A1, as follows) formed the clade of *Sclomina pallens* sp. nov.; the sister clade of *Sclomina pallens* sp. nov. contains the 11 SXYX samples of *Sclomina parva* sp. nov. distributed on Qingling Mountain of Shannxi province, the northernmost species of the genus. The 24 GZXR and 2 GZLS (GZLS1, GZLS6) samples that form a strongly supported clade, *Sclomina xingrensis* sp. nov., have the sister relationship with the clades of *S. pallens* sp. nov. and *S. parva* sp. nov. The other remaining 32 GZLS samples belonged to the large clade of *Sclomina erinacea* Stål, 1861 in BI and ML phylogeny trees; the result is consistent with the morphological characteristics and ABGD partition (Figure 1a,b, clade A). The two sympatric species of *Sclomina* occur together in Fangxiang county, Leishan town, Guizhou province (GZLS), which created confusion regarding the morphological identification in the past (GZLS1, 6), but now the DNA barcoding molecular data in this study resolved this issue. Within the spine-shaped connexivum groups, all the remaining samples (except the above GXLZ, GXTL, GZAL, GZXR, and SXYX samples, and GZLS1 and GLLS6) belonged to the clade of *Sclomina erinacea,* the spine-shaped connexivum group (Figure 1a,b, clade A). 

### 3.2. Species Delimitation

The overall mean divergence for the total dataset was 3.09%. The K2P sequence divergence between the 307 individuals ranged from 0–10.53% (Appendix A), between the 30 sampling groups from 0–10.53% (Appendix A), and between the pre-delimited species from 5.62–10.53% (Appendix A). Between individuals within the pre-delimited species, sequence divergence ranged from 0–0.97% (Appendix A), and between individuals within the 30 sampling groups, it ranged from 0–2.08% (Appendix A).

The ABGD analysis produced seventeen kinds of partition results when the prior maximal distance was 0.001000–0.048329 (1 group, 0.048329; 2 groups, 0.037927; 5 groups, 0.029764, 0.023357, 0.018330, 0.014384, 0.011288; 7 groups, 0.008859, 0.006952; 12 groups, 0.005456; 18 groups, 0.004281, 0.003360, 0.002637, 0.002069, 0.001624; 102 groups, 0.001274, 0.001000). We selected and accepted the partition result of 5 groups among the several results of ABGD when the prior maximal distance was 0.011288–0.029764 by comparing interspecific pairwise genetic distances (Appendix A), morphological identification (Figure 2 and Figure 3), and geographical information (Figure 1b). The five MOTUs delimited by AGBD were also well supported by NJ, BI, and ML trees.

### 3.3. Estimation of Divergence Time

The estimated divergence time showed that the different lineages in the genus *Sclomina* began to differentiate in the late Miocene of the Cenozoic Neogene and gradually formed in the Quaternary glacial Pleistocene (Figure 4). The most recent common ancestor (MRCA) of the genus *Sclomina* appeared at the end of the Miocene in the Cenozoic Neogene, 6.58 million years ago (Mya) with a 95% highest probability density (HPD) of 2.94–12.67 Mya. The first differentiation of *Sclomina guangxiensis* (clade E) occurred in the late Pleistocene of Quaternary glaciation, at 0.07 Mya (95% HPD: 0.01–0.25 Mya). The MRCA of the remaining clades was 3.46 Mya (95% HPD: 1.85–6.23 Mya). The MRCA of clades B, C, and D appeared at the end of the Pliocene, at 2.68 Mya (95% HPD: 1.32–4.70 Mya). Clade D was formed in the late Pleistocene at 0.25 Mya (95% HPD: 0.07–0.60 Mya). The MRCA of clades B and C appeared in the middle of the Pleistocene, at 1.55 Mya (95% HPD: 0.62–2.82 Mya). Clade B formed in the late Pleistocene at 0.64 Mya (95% HPD: 0.24–1.26 Mya). Clade C most recently formed in the late Pleistocene at 0.10 Mya (95% HPD: 0.02–0.30 Mya).

### 3.4. Biology

#### 3.4.1. Life History

We documented the annual life history of *Sclomina erinacea* based on the live specimens distributed in Fangxiang, Leishan, Guizhou (Appendix A). They occur one generation per year and overwinter as adults in the bush in Leishan in Guizhou province from early November to early April the following year. The overwintering adults lay eggs from early April to early June. One female can lay an average of 30 eggs in her whole lifetime. The egg stage is 16.93 ± 1.69 days (Appendix A). They lay eggs on the surface of leaves and stems of plants in a single-production manner (Figure 5a). There are five instars of nymphs (Figure 5b–i). Nymph stage is 108.6 ± 7.23 days from late April to early October (Appendix A). The adults emerge from early August to early October and peak in September. Then, the adults overwinter starting early November and begin to reproduce in early April to early June the following year. The adults may mate after emergence, but they cannot lay eggs until April of the following year. The adults generally die after oviposition in June. The longest longevity of an adult is about 300 days [35]. 

#### 3.4.2. Egg and Nymph

The nymph is green from second to fifth instars (Figure 5). So far, no green-bodied nymph has been found in the Reduviidae family. The coloration and structure of nymphs are similar to those of the symbiotic plant on which they grow, which is more conducive to survival. 

The egg is 2.00 mm in length, 1.00 mm in width, and oblong, with the middle part concaved. The egg gradually changes from yellowish brown to reddish brown, with a metallic shine, operculum is white (Figure 5a). 

The first instar nymph has a body that is pale yellowish brown to reddish brown, femora with dark brown tubers, tibiae that are pale greyish brown, with the basal part of tibiae having blackish brown annular markings, and the apical part of the tarsus is black; first to third antennal segments and basal part of the fourth are blackish brown (except for two annulations in the middle part of the first segment, one annulation in the middle part of the second, which are white to yellowish), most of fourth segment is reddish brown (except the basal part); the eyes are red. The body is covered with pale longer setae; the first antennal segment and the basal half of the second are sparsely covered with longer setae, the fourth segments with yellowish brown short setae (except the basal part); legs are sparsely covered with yellowish brown setae of various lengths. Head, thorax, femora, and abdomen dorsally have spines and tubers; the first antennal segment has several small tubercles. Body length is 2.00–2.50 mm (Figure 5b). 

The second instar nymph is similar to the first instar in coloration and structure, except the basal part of the abdomen is green, and the spines and tubers on the surface of the body become more distinct and thickened. Wing pad differentiation is not distinct. Body length is 3.30–3.42 mm (Figure 5c,d). 

The third instar nymph is similar to the first instar in coloration and structure, except most of the abdomen and most of the thorax become green, and spines and tubers on the surface of the body are distinct and thickened compared to those of the second instar. The wing pad is differentiated and very short. Body length is 7.10–8.00 mm (Figure 5e,f). 

The fourth instar nymph is similar to the third instar in coloration and structure, but most of the body has become green, and beneath the abdomen has white marking; spines and tubers on the surface of the body are more distinct and thickened than those of the third instar. The wing pad is longer, and the body length 9.80–10.50 mm (Figure 5g,h). 

The fifth instar nymph is totally green to yellowish green except for some markings on the antennae and legs similar to other instar nymphs; the middle part of the ventral side of the abdomen is milk-white; the spines and tubers on the surface of the head, thorax, abdomen, connexivum are very distinct and thickened compared to those of the fourth instar. The wing pad is distinct and long; body length is 12.00–12.30 mm (Figure 5i).

#### 3.4.3. Symbiotic Plants

The assassin bugs of the genus *Sclomina* prefer to inhabit the underbrush, mainly the *Rubus* species (Rosales: Rosaceae) and other plants that are armed with many spines and glandular hairs. In the present study, we recorded 22 species of symbiotic plants in 11 genera, 10 families, and eight orders in 14 sampling sites of the *Sclomina* species, as shown in Table A1. *Rubus* species are the main symbiotic plants (Figure 6a–o); the other plants are close to *Rubus* species. Assassin bugs are predatory and prey on other insects on the *Rubus* plant.

Most *Sclomina* species only inhabit symbiotic *Rubus* plants (Rosales: Rosaceae) (Table A1), such as *Sclomina parva* sp. nov. with *Rubus coreanus* (SXYX) (Figure 6g–i), *Sclomina guangxiensis* with *Rubus lambertianus* and *Rubus amphidasys* (GXLZ) (Figure 6d–f), *Sclomina xingrensis* sp. nov. with *Rubus lambertianus* (GZXR) (Figure 6j–l), and *Sclomina pallens* sp. nov. with *Rubus lambertianu*, *Rubus rosifolius* and *Rubus tephrodes* (GXAL and GZTL) (Figure 6m–o). However, the composition of symbiotic plants species is complicate in the different geographical groups of *Sclomina erinacea* (Figure 6a–c). The four sampling groups of *Sclomina erinacea*, GZQY, ZJZC, GDMZ and GZSB only live with *Rubus* spp. (Rosaceae) (Table A1). The group GXYL individuals were only found to inhabit *Salanum aculeatissimum* (Solanaceae), which is armed with long spines on its leaves, although there were many species of *Rubus* nearby. The group JXJG mainly lived on *Rubus alceifolius,* and few on *Rubus lambertianus* and *Rubus rosifolius*, with some adults on surrounding *Sambucus chinensis* (Adoxaceae), *Boehmeria nivea* (Urticaceae), *Woodwardia prolifera* (Blechnaceae), *Hibiscus mutabilis* (Malvaceae), and *Artemisia anomala* (Asteraceae). The group JXNC individuals were mainly found on *Rubus rosifolius*, some on *Macrothelypteris* sp. (Thelypteridaceae), *Pilea sinofasciata* (Urticaceae), and *Polygonum senticosum* (Polygonaceae). The group GZJP was found to live with *Rubus setchuenensis, Rubus rosifolius,* and *Polygonum senticosum* (Polygonaceae). The group GZLS lived on *Rubus rosifolius* and *Rubus lambertianus*, but nymphs preferred to occur on *Rhododendorn rivulare* (Ericaceae). 

### 3.5. Taxonomy

Order Hemiptera Linnaeus, 1735

Suborder Heteroptera Latreille, 1810

Family Reduviidae Latreille, 1807

Subfamily Harpactorinae Amyot and Serville, 1843


**Genus *Sclomina* Stål, 1861**


*Sclomina* Stål, 1861: 137; Maldonado-Capriles, 1990: 294; Putshkov, and Putshkov, 1996: 254; Hsiao, and Ren, 1981: 490. Type species by monotypy: *Sclomina erinacea* Stål, 1861.

Diagnostic characters: Body brown in adult, green in nymph, oblong ellique; body covered with numerous long spines and short tubercles. Head dorsally with two pairs of long spines behind antennal base and a pair of short spines behind ocelli; anteclypeus medial with two long spines and maxillary lateral, each with a short spine; first visible rostral segment sub-equal to one-half of second segment; first antennal segment sub-equal to head and pronotum together in length. Anterior angles of pronotum short spine-shaped; anterior pronotal lobe with four pairs of spines, posterior lobe with four strong spines; pro-pleuron of thorax and lateral margin of pronotum, Y-shaped ridge of scutellum with some little tubercles; pro-sternum of thorax lateral, each with a pair of short spines. Femora armed with spines and tubers; fore femur thickened, fore tibiae shorter than fore femur. Connexivum of abdomen laterally expended, posterior lateral angle obviously laterally produced, especially fourth to sixth segments. 

Distribution: China, Vietnam.


**Key to species of *Sclomina* Stål, 1861**


1.Body length more than 14 mm; post-lateral angle of abdominal connexivum spine-shaped laterally produced (Figure 3a–p)—2

-Body length less than 14 mm; post-lateral angle of abdominal connexivum Y-shaped laterally produced (Figure 3q–t)—3

2.Body yellowish-brown; spines on surface of head and pronotum not as strong; apical part of post-lateral angle of abdominal connexivum short spine-shaped (Figure 3a–o)—*Sclomina erinacea* Stål, 1861.

-Body dark reddish-brown; spines on surface of head and pronotum long, slender, and strong; post-lateral angle of abdominal connexivum long spine-shaped (Figure 3p)—*Sclomina guangxiensis* Ren, 2001.

3.Body length less than 13 mm; apical part of endosoma without spines (Figure 3t)—*Sclomina parva* sp. nov.

-Body length more than 13 mm but less than 14 mm; apical part of endosoma lateral, with a pair of large spines and three to five pairs of small spines (Figure 3q–s)—4

4.Two processes of post-lateral angle of connexivum somewhat acute; endosoma apically, with a pair of larger spines, and subapically with five smaller spines on one side and four smaller spines on the other side (Figure 3q,r)—*Sclomina pallens* sp. nov.

-Two processes of post-lateral angle of connexivum round; endosoma apical with a pair of larger spines and subapically lateral with four smaller spines on one side and three or four smaller spines on the other side (Figure 3s)—*Sclomina xingrensis* sp. nov.

#### 3.5.1. *Sclomina erinacea* Stål, 1861 

(Figure 2a–o, Figure 3a–o, Figure 5, Figure 6a–c and Figure 7)

*Sclomina erinacea* Stål, 1861: 137; Hsiao, and Ren, 1981: 490; Maldonado-Capriles, 1990: 294; Putshkov, and Putshkov, 1996: 254. 

Diagnosis: *S. erinacea* is the type species of the genus *Sclomina. S. erinacea* resembles *S. guangxiensis* in body shape and size. In the diagnosis section of *S. guangxiensis* in the following, we discuss the main distinguishing characteristics of the two species.

Redescription: Coloration: Body yellowish-brown to brown, with black markings (Figure 6a,b and Figure 7a–c). Structure: Post-lateral angle of connexivum gradually changeable in different geographical populations, as shown in Figure 2a–o. Pygophore elliptic, paramere clavate, apical part with thick setae (Figure 7d,e); phallosoma elliptic (Figure 7f–h). Number, shape, and size of spines of endosoma also variable in different geographical populations, nearly no spines or with very-small pale spines (Figure 3a–e), or six spines on one side and five spines on the other side (Figure 3f,g), or four pairs of spines (Figure 3h, Figure 7f–h), or five pairs of spines (Figure 3i–o).

Distribution: China (Anhui, Chongqing, Fujian, Guangdong, Guangxi, Guizhou, Hainan, Hunan, Jiangxi, Sichuan, Taiwan, Zhejiang); Vietnam (Phu Tho).

Remarks: The shape of the connexivum and the number and the size of spines on the apical part of the endosoma, which could provide morphological evidence for species division of *Sclomina* (as described in the key above), are also obviously variable in different geographical populations of *Sclomina erinacea* (Figure 2a–o and Figure 3a–o). We tried to show the complicated differences with pictures in Figure 2a–o and Figure 3a–o.

The lateral angle of the abdominal connexivum of *Sclomina erinacea* is angle-shaped and apical part is sharp (Figure 2a–o). However, the lateral angle of the connexivum in several groups from the southeast area of China (Figure 2a–e; JXNC, JXJG, ZJZC, FJQZ, GDQY samples; including GDMZ) is shorter than those in other populations, and the lateral angle is longer from the two island provinces of Taiwan and Hainan (TWNT and HNBW; Figure 2n,o).

Mostly, there are ten spines in the endosoma of the genitalia of *Sclomina erinacea* (Figure 3i–o; GZLS, GDSG, GXYL, GXJX, GZJP, TWNT, HNBW samples from Central China, and Taiwan and Hainan Islands). However, the endosoma in GZLP and GZHP is armed with eight spines (Figure 3h and Figure 7f–h). The spines of the endosoma are pale and small in several populations from the southeast area of China (Figure 3a–e; JXNC, JXJG, ZJZC, FJQZ, and GDQY samples, including GDMZ). The endosoma of AHFY and CQSMS samples is armed with eleven spines (Figure 3f,g). 

However, the genetic distance between individuals and populations of *Sclomina erinacea* was not significant compared to the five delimited species (Appendix A). The population change of this species deserves further analysis.

Symbiotic plants: *Rubus alceifolius*, *Rubus buergeri*, *Rubus lambertianus*, *Rubus rosifolius*, *Rubus setchuenensis*, *Rubus tsangorum*, *Rubus chiliadenus*, *Rubus tsangii*, *Rubus tephrodes*. Other surrounding symbiotic plants: *Boehmeria nivea*, *Hibiscus mutabilis*, *Macrothelypteris* sp., *Pilea sinofasciata*, *Polygonum senticosum*, *Rhododendron rivulare*, *Sambucus chinensis*, *Solanum aculeatissimum*, *Woodwardia prolifera*, *Artemisia anomala*.

Note: An expanded description of *Sclomina erinacea* Stål, 1861 is provided in Appendix A of the Appendix A.

#### 3.5.2. *Sclomina guangxiensis* Ren, 2001

(Figure 2p, Figure 3p, Figure 6d–f and Figure 8)

*Sclomina guangxiensis* Ren, 2001: 1.

Diagnosis: This species *S. guangxiensis* is similar to *S. erinacea*, but in the former the body color is dark reddish-brown, the posterior pronotal lobe is with black markings, and the post-lateral angles of the abdominal connexiva are long and narrow (vs. in *S. erinacea,* the body color is brown to yellowish brown, the posterior pronotal lobe is without black markings, and the post-lateral angles of the abdominal connexiva are shorter).

Redescription: Body coloration: Body dark reddish-brown with black and yellow markings (Figure 6d,e and Figure 8a–c). Structure: Two long spines on the middle part of the posterior pronotal lobe bent backward, two spines on the middle-posterior part of the anterior lobe bent forward (Figure 8b); post-lateral angles of abdominal connexival segments long spine-shaped produced laterally (Figure 2p and Figure 8a–c). Pygophore elliptic, median pygophore process medianly concave (Figure 8d,e); paramere clavate, apical part with thick setae (Figure 8f); apical part of endosoma with two pair of large spines in middle part, and laterally with three spines on one side and four spines on the other side (Figure 3p and Figure 8g–j).

Type material examined: Holotype: male, Guangxi, Daxin, 2800 m, 1998-III-19, Li Wenzhu leg., kept in NKU. Paratypes: 1 female, Guangxi, Ningming, Longrui National Natural Reserve, 1984-V-22, 200 m, Ren Shuzhi leg., kept in NKU; 1 female, Guangxi, Ningming, Longrui National Natural Reserve, 1984-V-19, Ren Shuzhi leg., kept in NKU.

Other material examined: China, Guangxi: 20 males, 20 females (CAU, SG-GXLZ1–40), Guangxi, Longzhou, Nonggang, 2017-VIII-22, 2017-XI-11, 2020-IV-30, Zhao Ping and Yang Mingyuan leg.; 1 male, 2 females (CAU, SG-GXLZ41–43), Guangxi, Longzhou, Nonggang, 2006-V-13, Huang Xia and Shi Zhongting leg.; 1 male, 1 female (CAU, SG-GXLZ44–45), Guangxi, Longzhou, Nonggang, 2006-X-4, Huang Xia leg.

Distribution: China (Guangxi).

Remarks: In terms of the morphological evidence and the geographical distribution (Figure 1b, Figure 2 and Figure 3), we conjectured at first that *Sclomina guangxiensis* is a special case of the *Sclomina erinacea* cline and we seriously doubted the taxonomic status as a valid species. Through the integrative taxonomy based on DNA barcoding and morphological evidence, the result showed that *Sclomina guangxiensis* is a valid species and is a far relative species with the largest genetic distance from other members of *Sclomina* (Figure 1a,c and Figure 4).

Symbiotic plants: Rubus amphidasys and Rubus lambertianus.

Note: An expanded description of *Sclomina guangxiensis* Ren, 2001 is provided in Appendix A.

In addition, about the type specimens, Ren (2001) set up the species *Sclomina guangxiensis* Ren, 2001 based on one male and two female specimens and assigned the female specimen (Guangxi, Ningming, Longrui National Natural Reserve, 1984-V-22) as Holotype in the paper [6]. However, when we checked the type specimen kept in NKU, we found that the Holotype red label is under the male specimen (Guangxi, Daxin, 1998-III-19). It can be seen from her paper that most of the description and illustration of the species was based on this male specimen, so we chose this male specimen as Holotype herein. The Holotype red label indicates that the type specimens are deposited in the Institute of Zoology, Chinese Academy of Sciences, but they are actually deposited in Nankai University (NKU).

#### 3.5.3. *Sclomina pallens* P. Zhao and Cai sp. nov.

http://www.zoobank.org/urn:lsid:zoobank.org:act:6BC7BA8A-DB6A-4E22-930A-B22BE7FF288A.

(Figure 2q,r, Figure 3q,r, Figure 6m–o and Figure 9) 

Diagnosis: The Y-shaped connexivum group of *Sclomina* consists of three new cryptic species: *Sclomina pallens* sp. nov., *Sclomina parva* sp. nov., and *Sclomina xingrensis* sp. nov. (Figure 1, Figure 2q–t, Figure 3q–t and Figure 4), which are easily distinguished with other two known species, *S. erinacea* and *S. guangxiensis* by the Y-shaped post-lateral angles of connexiva, and the smaller spines of endosoma (Figure 2 and Figure 3) (vs. in *S. erinacea* and *S. guangxiensis*, the post-lateral angles of connexiva are spine-shaped, and the spines of endosoma are larger).

*S. pallens* sp. nov. is most similar to *S. xingrensis* sp. nov. in general body shape and size. In the diagnosis section of *S. xingrensis* sp. nov. in the following, we discuss the main distinguishing characteristics of the two species.

Description: Body coloration: Body pale yellowish-brown with black and brown markings (Figure 6m,n and Figure 9a–f). Antennae reddish-brown; apical part of first segment, apical part of second, and basal part of third black; two median annulations of first antennal segment, subapical annulation of second segment, sub-basal part of third segment paler, pale yellowish-brown; second segment (except apical and subapical parts), third segment (except basal part), fourth segment yellowish-brown. Longitudinal strips of head ventrally and laterally, markings of transverse constriction of pronotum, longitudinal stripes of fore femur, markings of coxae and trochanters, two oblique strips of basal part of tibiae, markings of propleural episternum, markings of meso- and meta-pleuron of thorax, markings on lateral side of abdomen, blackish-brown to black; three obscure annulations of femur brown; membrane of fore wing transparent, vein pale brown (except basal part yellowish to milk-white), middle part of corium with small black markings; basal half and posterior margin of connexivum pale yellowish-brown, posterior half and posterior-lateral angle of each connexival segment brown; middle part of abdomen dull yellow, lateral side with black and yellow markings. Structure: Body median-sized. Body covered with curved depressed short yellowish setae and sparsely covered with sub-vertical long setae. First antennal segment longest, apical part bent; spines on head and pronotum shorter; two spines at the middle part of the posterior pronotal lobe slightly bent backward, two spines on middle-posterior part of anterior lobe slightly bent forward (Figure 6m,n and Figure 9a–f); post-lateral angle of abdominal connexivum Y-shaped produced laterally, slightly acute (Figure 2q,r, Figure 6m,n and Figure 9b,e). Pygophore elliptic, median pygophore process slightly medianly concave (Figure 9g); paramere clavate, apical half with setae (Figure 9g,h); dorsal phallothecal sclerite sclerotized and flat, tongue-shaped; endosoma apically lateral with a pair of large spines and subapically lateral with five spines (Figure 9l,n) and four spines on the other side (Figure 9k,o).

Measurements (males (*n* = 11)/females (*n* = 8), in mm): Body length 13.80 ± 0.30/13.59 ± 0.13; maximal width of abdomen 4.34 ± 0.40/4.47 ± 0.04. Head length 3.02 ± 0.03/2.97 ± 0.04; length of anteocular part 1.26 ± 0.06/1.23 ± 0.04; length of postocular part 1.28 ± 0.09/1.23 ± 0.04; length of synthlipsis 0.69 ± 0.03/0.72; distance between ocelli 0.58 ± 0.03/0.60; length of antennal segments I–IV = 5.96 ± 0.30/5.88: 2.97 ± 0.06/2.94: 1.75 ± 0.06/1.56: 2.71 ± 0.14/2.67; length of visible rostral segments I–III = 1.18 ± 0.03/1.20: 1.51 ± 0.10/1.44 ± 0.08: 0.44 ± 0.03/0.44 ± 0.02; length of pronotum 2.45 ± 0.07/2.52 ± 0.08; maximal width of pronotum 4.20 ± 0.16/4.29 ± 0.13; length of hemelytron 8.60 ± 0.69/8.49 ± 0.13.

Type specimens: Holotype: male (CAU, SP-GZAL23), China, Guizhou, Anlong, Xianheping, 2013-X-12–15, Zhao Ping and Yang Kun leg. Paratypes: 7 males, 15 females (CAU, SP-GZAL1–22), China, Guizhou, Anlong, Xianheping, 2017-X-1, 2018-VII-25, 2018-IX-8, Zhao Ping and Luo Maomao leg.; 10 males, 13 females (CAU, SP-GZTL1–23), Guangxi, Tianlin, Langping, 2017-VIII-22, Zhao Ping leg.

Etymology: The new species is named after the pale body color.

Distribution: China (Guangxi, Guizhou).

Symbiotic plants: *Rubus lambertianu*, *Rubus rosifolius*, and *Rubus tephrodes*.

#### 3.5.4. *Sclomina parva* P. Zhao and Cai sp. nov. 

http://www.zoobank.org/urn:lsid:zoobank.org:act:4D574F21-64D9-4890-81DB-CAE9223A741E.

(Figure 2t, Figure 3t, Figure 6g–i and Figure 10)

Diagnosis: The new species is similar to *Sclomina pallens* P. Zhao and Cai sp. nov. in that the post-lateral angles of abdominal connexivum are Y-shaped, slightly acute, apical part is blunt (Figure 2t and Figure 10a–c). But the former is easily distinguished from the latter and other three species of the genus by following characters: there are no spinous processes at apical part of endosoma of the male genitalia (Figure 3t and Figure 10f–h); the body size is smallest in the genus; in addition, it is the northernmost species of the genus (vs. in *S. pallens* sp. nov. and other three species, the endosoma is armed with spines, and the body size is larger).

Description: Body coloration: Body pale yellowish-brown with blackish-brown markings (Figure 6g,h and Figure 10a–c). Longitudinal strips of head lateral and posterior lobe, anterior-lateral markings of posterior pronotal lobe, markings of coxae and tranchanters, markings of propleural episternum, markings of meso- and metapleuron of thorax, markings on lateral side of abdomen blackish-brown to black; first antennal segment reddish-brown, two median annulations yellowish-white; second segment pale brown, subapical annulation yellowish-white, apical part black; sub-basal part of third segment yellowish-white, basal part black, and apical half pale brown; fourth segment pale brown. Femora yellowish-brown, three obscure annulations brown; tibiae pale yellowish-brown, basal part with two dark brown oblique stripes, apex yellowish-brown; tarsus blackish-brown. Membrane of fore wing transparent, vein brown (except basal vein milk-white), middle part of corium, clavus brown; basal half and posterior margin of abdominal connexivum pale yellowish-brown, posterior-lateral angle of each connexival segment dark brown; middle part of abdomen dull yellow, lateral side with black and brown markings. Structure: Body smaller. Body covered with white depressed short setae. First antennal segment longest, apical part bent. Spines behind antennal base bent slightly forward; apical part of two spines on middle part of posterior pronotal lobe bent backward. Post-lateral angles of abdominal connexivum Y-shaped, apical part round (Figure 2t and Figure 10a–c). Pygophore elliptic, median pygophore process not concave in middle (Figure 10d); paramere clavate, apical half with thick setae, basal part bent (Figure 10d,e); phallosoma elliptic; dorsal phallothecal sclerite sclerotized and flat, tongue-shaped; endosoma without spines (Figure 3t and Figure 10f–h).

Measurements (males (*n* = 3), in mm): Body length 12.73 ± 0.03; maximal width of abdomen 4.47 ± 1.18. Head length 3.15 ± 0.1; length of anteocular part 1.13 ± 0.06; length of postocular part 1.31 ± 0.13; length of synthlipsis 0.67 ± 0.09; distance between ocelli 0.55 ± 0.01; length of antennal segments I–IV = 5.66 ± 0.65: 2.96 ± 0.28: 1.55 ± 0.39: 2.76 ± 0.68; length of visible rostral segments I–III = 1.06 ± 0.15: 1.43 ± 0.04: 0.44 ± 0.13; length of pronotum 2.47 ± 0.1; maximal width of pronotum 4.21 ± 0.32; length of hemelytron 8.05 ± 0.18.

Type specimens: Holotype: male (CAU, SPA-SXYX14), China, Shannxi, Hanzhong, Yangxian, Huayang, Hongjunlin, 2020-VIII-8, Zhao Ping and Yuan Meixuan leg. Paratypes: 1 male (CAU, SPA-SXYX15), China, Shannxi, Hanzhong, Yangxian, Huayang, Hongjunlin, 2020-VIII-8, Zhao Ping and Yuan Meixuan leg.; 5 nymphs (CAU, SPA-SXYX1–5), China, Shannxi, Hanzhong, Yangxian, Huayang, Hongjunlin, 2017-VIII-5; 1 male, 7 nymphs (CAU, SPA-SXYX6–13), China, Shannxi, Hanzhong, Yangxian, Huayang, Hongjun, 2018-VIII-6, Zhao Ping and Li Donghai leg.

Etymology: The specific name refers to the small body size of the new species.

Female: Unknown.

Distribution: China (Shannxi).

Symbiotic plant: *Rubus coreanus*.

#### 3.5.5. *Sclomina xingrensis* P. Zhao and Cai sp. nov. 

http://www.zoobank.org/urn:lsid:zoobank.org:act:AF0C8158-F774-49E6-85CE-CE2895F1AC11.

(Figure 2s, Figure 3s, Figure 6j–l and Figure 11) 

Diagnosis: *S. xingrensis* sp. nov. is so similar to *Sclomina pallens* sp. nov. in morphological characteristics, in that they are very difficult to distinguish, even though the genetic distance between the two species is distinctly larger (Appendix A). Through the anatomy and comparison of the male external genitalia, we found that the apical part the endosoma is armed with a pair of larger spines and the subapical part is armed with four spines on one side and three or four spines on the other side in *S. xingrensis* sp. nov. (Figure 3s and Figure 11i–n) (vs. in *S. pallens* sp. nov., the apical part of endosoma has a pair of larger spines and the subapical part has five spines on one side and four spines on the other side, as shown in Figure 3q,r and Figure 9k,l,n,o). In addition, the body color of *S. pallens* sp. nov. is paler than that of *S. xingrensis* sp. nov., and the structure of abdominal connexivum is slightly different.

Description: Body coloration: Body pale brown with brown and black markings (Figure 6j,k and Figure 11a–f). Antennae brown, subbasal and subapical annulations of first segment paler, subapical annulation of second segment paler and apical part black, basal part of third segment black and subbasal part pale yellowish-brown; longitudinal strips of head ventral and lateral, longitudinal strips of fore femur, three annulations of mid- and hind-femur, two subbasal annulations of tibiae, markings of coxae, propleural episternum, markings of meso- and metapleuron of thorax, markings on lateral side of abdomen, blackish-brown to black; membrane of fore wing transparent and pale brown, vein brown (except basal part yellowish to milk-white), middle part of corium black-brown; basal half of connexivum yellowish-brown, posterior margin of each connexival segment pale milk-white; abdomen ventrally milk white, lateral side with black and yellow markings. Structure: Body median-sized. Body covered with curved depressed short yellowish setae and sparsely clothed with subvertical long setae. First antennal segment longest, apical part bent; spines on head dorsally and pronotum shorter and thickened, and with several small tubers; two spines at the middle part of the posterior pronotal lobe bent backward; two spines on middle-posterior part of anterior lobe sub-vertical, apical part somewhat bent forward; post-lateral angle of abdominal connexivum Y-shaped produced laterally, apical part round (Figure 2s). Pygophore elliptic, median pygophore process medianly slightly concave; paramere clavate, apical half with setae (Figure 11g,h); dorsal phallothecal sclerite sclerotized and flat, tongue-shaped (Figure 11i,m); endosoma apically lateral with a pair of large spines and subapically lateral with four spines on one side and three or four spines on the other side (Figure 3s and Figure 11i–k, GZXR1, Figure 11l–m, GZXR2, Figure 11n, GZLS1).

Measurements (males (*n* = 7)/females (*n* = 3), in mm): Body length 13.47 ± 0.3/13.68 ± 0.24; maximal width of abdomen 4.10 ± 0.46/4.67 ± 0.30. Head length 2.94 ± 0.12/2.96 ± 0.18; length of anteocular part 1.24 ± 0.06/1.33 ± 0.14; length of postocular part 1.21 ± 0.11/1.29 ± 0.12; length of synthlipsis 0.73 ± 0.03/0.72; distance between ocelli 0.56 ±0.03/0.62±0.03; length of antennal segments I–IV= 5.32 ± 0.28/5.15: 2.80 ± 0.11/2.72: 1.56/1.56: 2.67/2.8; length of visible rostral segments I–III = 1.14/1.09 ± 0.07: 1.43 ± 0.16/1.57 ± 0.12: 0.50 ± 0.07/0.48 ± 0.07; length of pronotum 2.36 ± 0.07/2.48 ± 0.18; maximal width of pronotum 4.37 ± 0.33/4.53 ± 0.07; length of hemelytron 8.60 ± 0.40/9.20 ± 0.18.

Type specimens: Holotype: male (CAU, SX-GZXR42), China, Guizhou, Xingren, Lishuping, 2020-XIII-15, Zhao Ping leg., kept in CAU. Paratypes: 6 males, 3 females (CAU, SX-GZXR1–9), 2018-VII-26, 15 males, 15 females, SX-GZXR10–39, 2018-IX-8, 1 male, 1 female, 2020-XIII-15, SX-GZXR40–41, Guizhou, Xingren, Lishuping, Zhao Ping, Yang Kun and Li Donghai leg.; 1 male, 2 nymphs (CAU, SX-GZLS1, SX-GZLS6), China, Guizhou, Leishan, Fangxiang, Pingxiang, 2018-VII-26, Zhao Ping and Li Donghai leg.

Etymology: The species name *xingrensis* alludes to the type of locality of the species in Xingren county, Guizhou province, China.

Distribution: China (Guizhou).

Symbiotic plant: *Rubus lambertianus*.

## 4. Discussion

### 4.1. Phylogeny and Species Delimitation 

In classical taxonomy, the adult’s external structures are the main characteristics used in the species identification of Reduviidae. The structure differences in the male external genitalia are the ultimate evidence of the closely related species division in Reduviidae. However, distinct morphological differences are sometimes difficult to detect, especially in some closely related species. In this study, we performed the phylogenetic analyses and species delimitation based on the DNA barcoding sequences of 307 *Sclomina* individuals from 30 sampling localities to recover three cryptic species and resolve the problem of morphological identification. All samples were divided into five clades and were identified as the following five species (Figure 1): *Sclomina erinacea* Stål, 1861; *Sclomina xingrensis* sp. nov.; *Sclomina pallens* sp. nov.; *Sclomina parva* sp. nov.; and *Sclomina guangxiensis* Ren, 2001. The molecular phylogeny effectively guided species identification based on morphological characteristics, showing that the *COI* DNA barcoding sequence is an effective method for identifying these closely related species. 

*Sclomina erinacea*, widely distributed in Southern China, shows some morphological divergence (Figure 2a–o and Figure 3a–o) and has a complicated relationship with many symbiotic plant species (Table A1). Although three cryptic species were detected in our study, there are still possible additional cryptic species in *Sclomina erinacea* populations. The island samples of TW, TWPD, TWNT, and HNBW distributed in Taiwan and Hainan Islands of China showed a relatively high divergence level between them and other samples from China mainland (Appendix A; Figure 1a, clade A; Appendix A, mean distances of TW, TWPD, and TWNT between groups is 1.46–3.40%; that of HNBW is 2.01–3.34%). The overall mean divergence for the total dataset is 3.09% and those of the delimited species ranges from 5.62% to 10.53% (Appendix A). Because these island populations showed no obvious phenotypic morphological divergence and phylogenetic divergence in the BI and ML trees, they are not proposed to be new species. A population genetic study of *Sclomina erinacea* based on more samples and gene sequences should be performed in a future study. 

### 4.2. Evolutionary History

Higher Reduviidae originated in the Middle Jurassic (178 Mya), but significant lineage diversification only began in the Late Cretaceous (97 Mya, 65–96 Mya), and those in reduviid subfamily Harpactorinae occurred in Miocene [36]. Results of our study suggest that the *Sclomina* began to differentiate in the late Miocene of the Cenozoic Neogene (Figure 4). It was speculated that the earliest differentiation of the genus occurred at about 6.58 Mya, the second at about 3.46 Mya, and the third at about 2.68 Mya (Figure 4), which were related to Orogeny in Western China due to the rise of the Himalayas in the late Miocene and Pliocene.

According to the maximum credibility tree of the divergence time and phylogenetic trees (ML and BI), the clade of *Sclomina guangxiensis* was suggested to diverged about 6.58 Mya and is quite far from the other members of *Sclomina.* Southwestern Guangxi province in China is a special area that was not directly invaded and destroyed by the Quaternary continental glaciers, and the biological development process was not interrupted, so many ancient organisms were preserved. So, the clade of *Sclomina guangxiensis* possibly formed in the phyletic speciation model without the influence of geographical isolation.

The divergence of these remaining clades A, B, C, and D (except clade E in Figure 4) of the genus should be related to allopatric speciation model with geographical isolation. The second divergence of *Sclomina*, the Y-shaped connexivum group, and the spine-shaped connexivum group began to split about 3.46 Mya (Figure 4). Due to the Orogenic movement, Western China was mountainous and the altitude was increased, but Eastern China was flat and lakes were widespread. Based on our phylogenetic study and the geographical pattern of the genus, there is a sharp species differentiation of the Y-shaped connexivum group (*Sclomina pallens* sp. nov., *Sclomina xingrensis* sp. nov., and *Sclomina parva* sp. nov.) along the edge of Western China’s mountains areas at altitudes of no more than 2000 m (Figure 1 and Figure 4) due to mountain rising [37]. However, the spine-shaped connexivum group, *Sclomina erinacea* Stål, 1861, which is widely distributed in Southern China, did not undergo speciation evolutionary differentiation, and the geographical populations always maintain mutual gene exchange, and the genetic differences are not significant due to the flat terrain in Eastern China and the absence of geographical isolation (Figure 1 and Figure 4; Appendix A). The genus is only distributed in Southern China and not beyond the Qinling Mountains. *Sclomina parva* sp. nov. is a unique species in the genus *Sclomina* due to having the smallest body and endosoma without any spines; it has the northernmost distribution (southwest of the Qinling Mountains range). The Qinling Mountains, which are the boundary between the Oriental and the Palearctic regions in China, produced an obvious blocking effect on the northward expansion of *Sclomina*. As such, speciation events of the genus may have occurred due to the end of the last glacial conditions in the Pleistocene and the warm and humid climate in the Holocene. Therefore, the five species gradually formed in the Pleistocene. 

### 4.3. Symbiotic Relationship between Sclomina and Rubus: Mimicry and Protective Color 

The occurrence of green coloration in phytophages; red, yellow, brown, or gray coloration in predators or phytophages; and the same coloration as flowers when nymphs feed or live on the flower is widespread in the Heteroptera [1]. Through field investigation, we found that all members in the reduviid genus *Sclomina* strictly inhabit the plant genus *Rubus* (Figure 6), except that *Sclomina erinacea* also exist on surrounding other plants. Huang et al. reported that *Sclomina erinacea* prefers to live with ferns in Hunan province, China [35]. We also found the bugs inhabited *Rubus* and its surrounding fern plants in JXNC and JXJG samples (Table A1). We recorded the 22 symbiotic plants species belonging to 11 genera and 10 families from 14 collection sites of the species representing the genus *Sclomina* (Table A1). *Rubus* spp. (Rosaceae: Rosaceae) are the dominant symbiotic plant species. These symbiotic plants are covered with abundant glandular hairs and spines. The predatory assassin bug genus *Sclomina* can survive and reproduce when they are raised in the laboratory after leaving the symbiotic plant, so we thought the relationship was symbiotic between *Sclomina* and *Rubus* plants. There is no report of symbiotic plants of the family Reduviidae, except the tribes Ectinoderini, Diaspidiini, and Apiomerini in the reduviid subfamily Harpactorinae inhabiting on the bark of pine tree, the fore tibiae of which are covered with plant resin to catch prey [38,39].

Mimicry is a kind of ecological adaptation in which one or both species benefit by imitating another living creature in the form and behavior characteristics. It is a special animal behavior formed through long-term evolution in nature because of predation pressure, reproduction, etc. [40,41]. The green nymphs and brown adults of the genus *Sclomina* are covered with many spines and setae on head, pronotum, and legs, and the connexivum of the abdomen is spine- or Y-shaped, produced laterally, which is similar to the inflorescences, stems, stipules, and leaves of symbiotic Rubus plants (Figure 5 and Figure 6). The protection color and mimicry of the assassin bug are adapted to the symbiotic plant on which they live. In addition, we found that the reduviid bugs only live on healthy and prosperous *Rubus* plants, but not on plants seriously damaged by pests. We deduced that the reduviid bugs may mimic the spines and glandular hairs structure of *Rubus* plant to gain habitat, protection, and food (Figure 6), while the host plant is protected from pests. The coloration and structure of adults and nymphs resemble those of the *Rubus* plant, which benefits both species. Even if the assassin bugs emerge and adults can fly, they will not be far away from the host plants. We deduced that the symbiotic relationship and the poor migration ability are two of the main reasons for the rapid genetic differentiation and distinct phenotypic divergence of *Sclomina*. The main origin of *Rubus* is thought to be Southwest China, where there are abundant species and extensive morphological variation of *Rubus* [42]. Similarly, the significant speciation and species differentiation of the genus *Sclomina* occur in Southwest China. Whether co-evolution occurred between the plant and bug requires future studies. 

## 5. Conclusions

In the present study, we conducted species delimitation and phylogenetic analyses based on the DNA barcodes of 307 *Sclomina* specimens collected from 30 sampling locations. All samples in this study were identified as five species, among which *Sclomina guangxiensis* Ren, 2001 is a valid species, and the Chinese common reduviid *Sclomina erinacea* Stål, 1861 actually includes three new cryptic species: *Sclomina xingrensis* P. Zhao and Cai sp. nov., *Sclomina pallens* P. Zhao and Cai sp. nov., and *Sclomina parva* P. Zhao and Cai sp. nov. The integrative taxonomy based on DNA barcoding and morphological evidence was necessary and effective for accurate identification of the genus *Sclomina*. The biological information of *Sclomina erinacea* and the symbiotic relationships between Sclomina and Rubus plants were reported, which provided new insights to the spinous reduviid genus *Sclomina*. 

## Figures and Tables

**Figure 1 insects-12-00251-f001:**
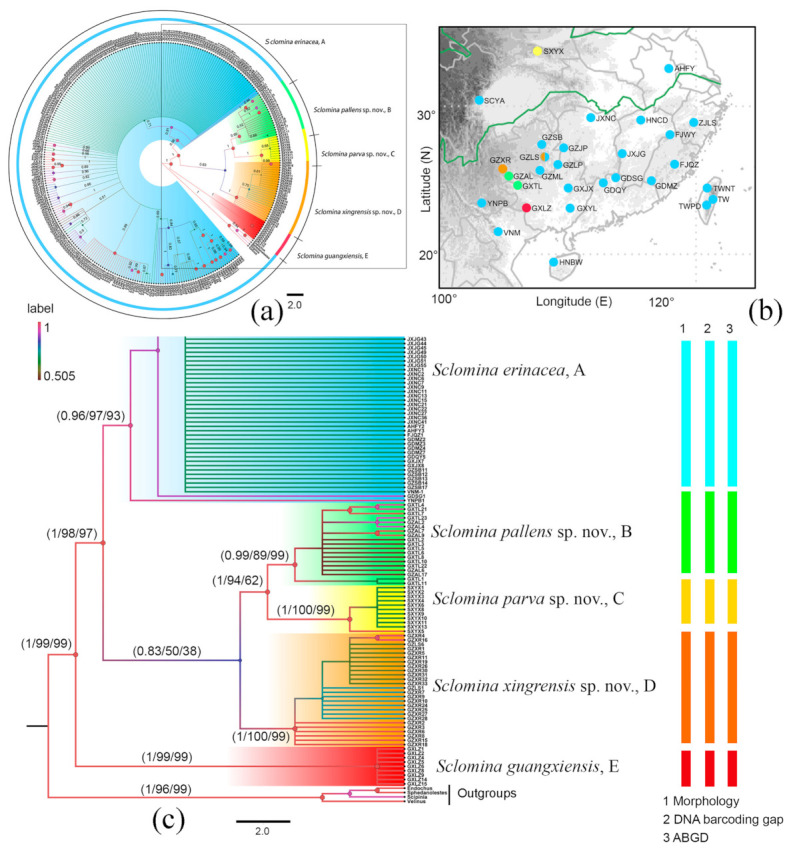
(**a**) Bayesian phylogenetic tree of cytochrome *c* oxidase subunit I (*COI*) sequence for 307 terminals of *Sclomina*; (**b**) enlarged box from (**a**); the results of Bayesian inference (BI), maximum likelihood (ML) and neighbor-joining (NJ) analyses agree with those of the three different species delimitation approaches, morphology identification, DNA barcoding gap and ABGD (automatic barcoding gap discovery) analysis. The numbers above the branches are the posterior probabilities of the Bayesian inference (BI), the bootstrap values of maximum likelihood (ML) analyses, and neighbor-joining (NJ). (**c**) Distribution map of the 5 *Sclomina* species and locality group terminals (for sample codes, see Table A1). The clades/species identified in this study are indicated in different colors: blue, *Sclomina erinacea* Stål, 1861, clade A; green, *Sclomina pallens* sp. nov., clade B; yellow, *Sclomina parva* sp. nov., clade C; orange, *Sclomina xingrensis* sp. nov., clade D; red, *Sclomina guangxiensis* Ren, 2001, clade E. Legend label, the Bayesian posterior probability (BPP) of Figure 1a,c.

**Figure 2 insects-12-00251-f002:**
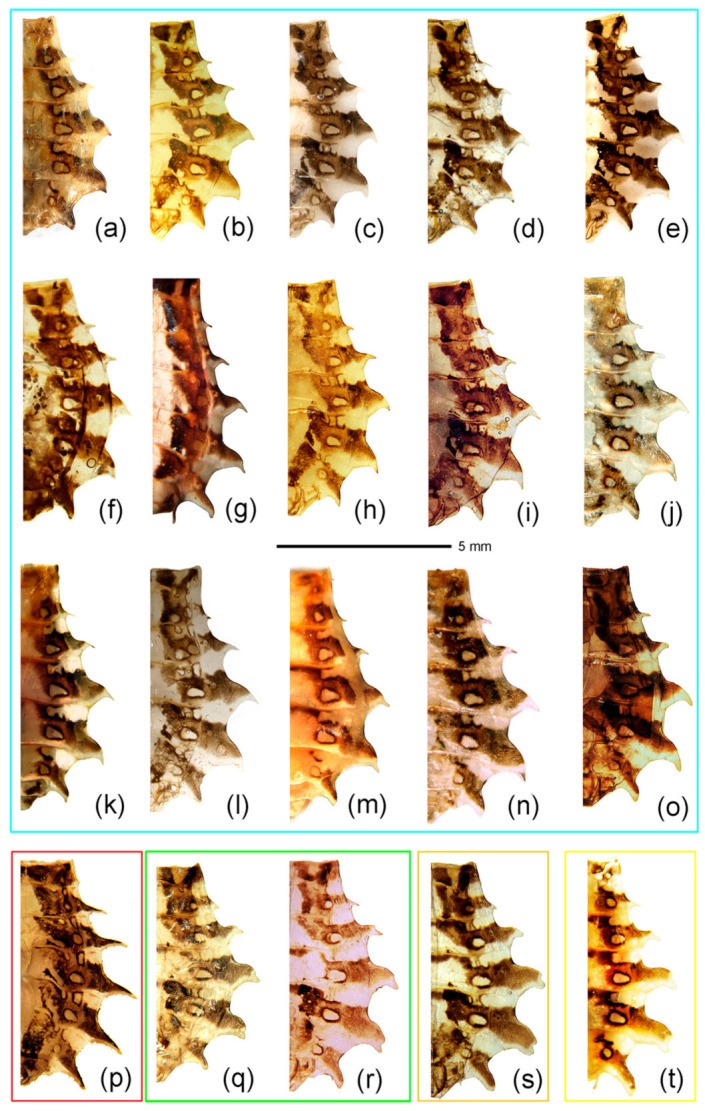
Half of abdomen showing the structural changes in the post-lateral angle of the connexivum. (**a**–**o**) *Sclomina erinacea* Stål, 1861; (**p**) *Sclomina guangxiensis* Ren, 2001; (**q**,**r**) *Sclomina pallens* sp. nov.; (**s**) *Sclomina xingrensis* sp. nov.; (**t**) *Sclomina parva* sp. nov. (**a**) JXNC; (**b**) JXJG; (**c**) ZJZC; (**d**) FJQZ; (**e**) GDQY; (**f**) AHFY; (**g**) CQSMS; (**h**) GZLP; (**i**) GZLS; (**j**) GDSG; (**k**) GXYL; (**l**) GXJX; (**m**) GZJP; (**n**) TWNT; (**o**) HNBW; (**p**) GXLZ; (**q**) GXTL; (**r**) GZAL; (**s**) GZXR; (**t**) SXYX (for sample codes, see Table A1). Scale bar = 5 mm.

**Figure 3 insects-12-00251-f003:**
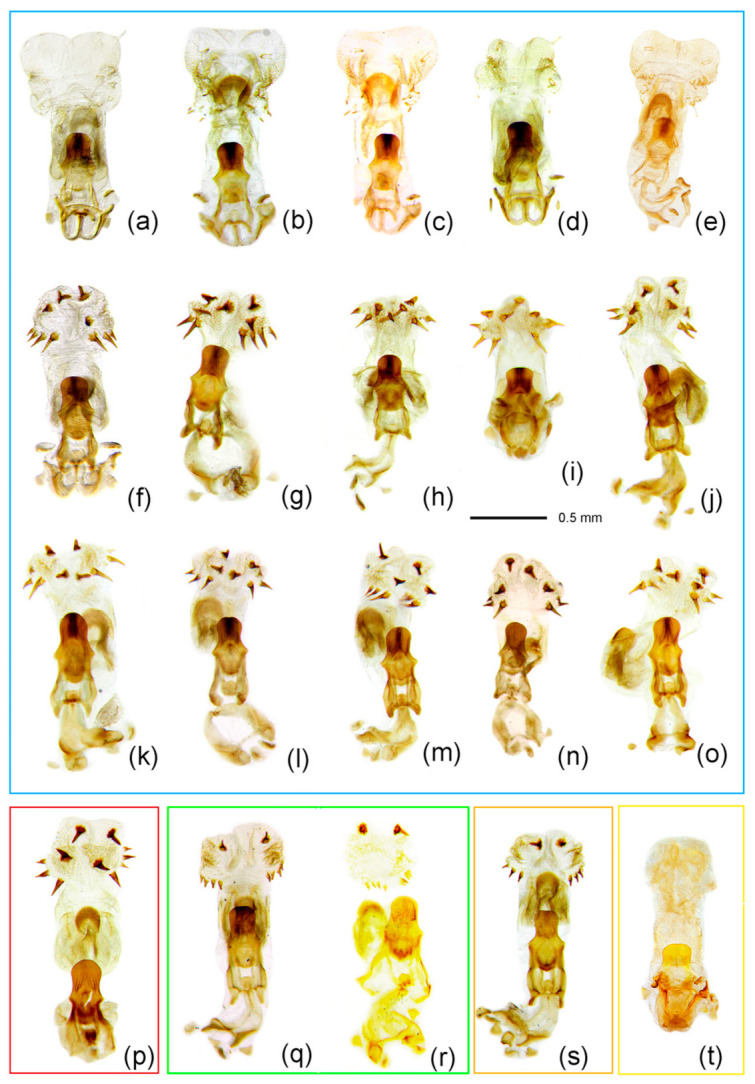
Phallus; the number and size variability of spines on the apical part of endosoma. (**a**–**o**) *Sclomina erinacea* Stål, 1861; (**p**) *Sclomina guangxiensis* Ren, 2001; (**q**,**r**) *Sclomina pallens* sp. nov.; (**s**) *Sclomina xingrensis* sp. nov.; (**t**) *Sclomina parva* sp. nov. (**a**) JXNC; (**b**) JXJG; (**c**) ZJZC; (**d**) FJQZ; (**e**) GDQY; (**f**) AHFY; (**g**) CQSMS; (**h**) GZLP; (**i**) GZLS; (**j**) GDSG; (**k**) GXYL; (**l**) GXJX; (**m**) GZJP; (**n**) TWNT; (**o**) HNBW; (**p**) GXLZ; (**q**) GXTL; (**r**) GZAL; (**s**) GZXR; (**t**) SXYX (for sample codes, see Table A1). Scale bar = 0.5 mm.

**Figure 4 insects-12-00251-f004:**
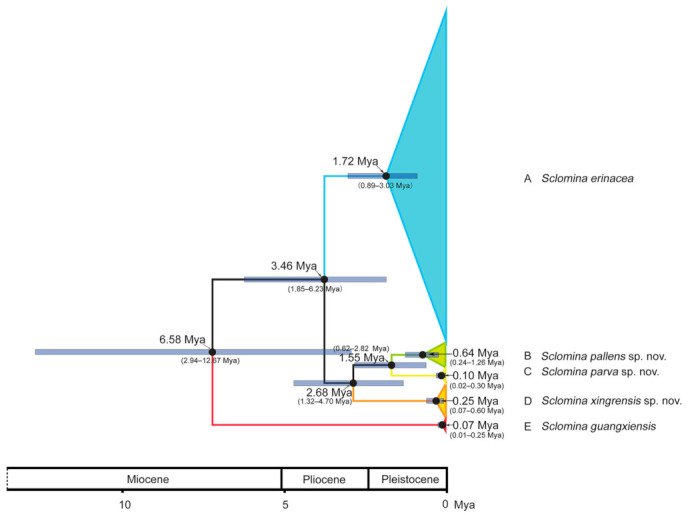
Divergence time tree based on *COI* data. The clades/species identified in this study are indicated in using the same colors as in Figure 1. The 95% highest posterior density (HPD) intervals are represented by pale blue bars. The *x*-axis depicts the corresponding geological periods. Mya: million years ago.

**Figure 5 insects-12-00251-f005:**
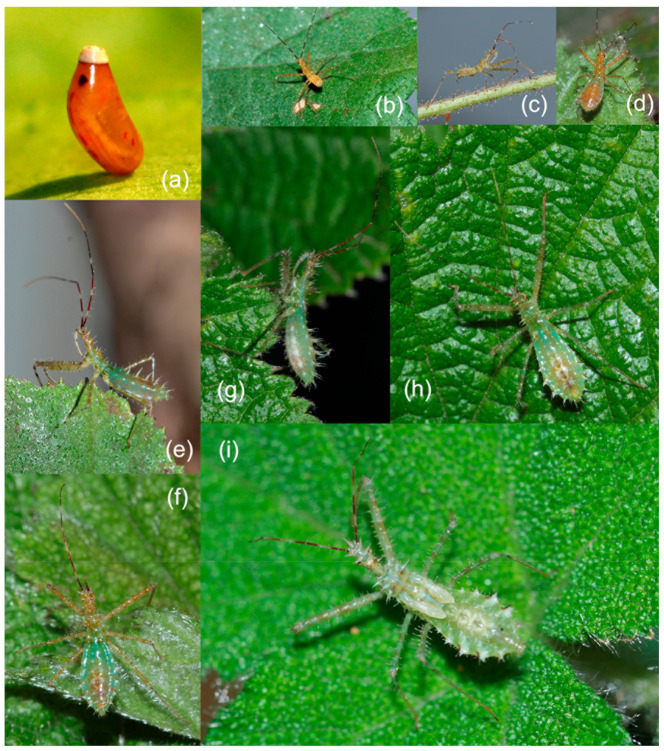
*Sclomina erinacea* Stål, 1861, GZLS. (**a**) egg; (**b**–**i**) nymph. (**a**) reddish brown egg before hatching; (**b**) first instar; (**c**,**d**) second instar; (**e**,**f**) third instar; (**g**,**h**) fourth instar; (**i**) fifth instar.

**Figure 6 insects-12-00251-f006:**
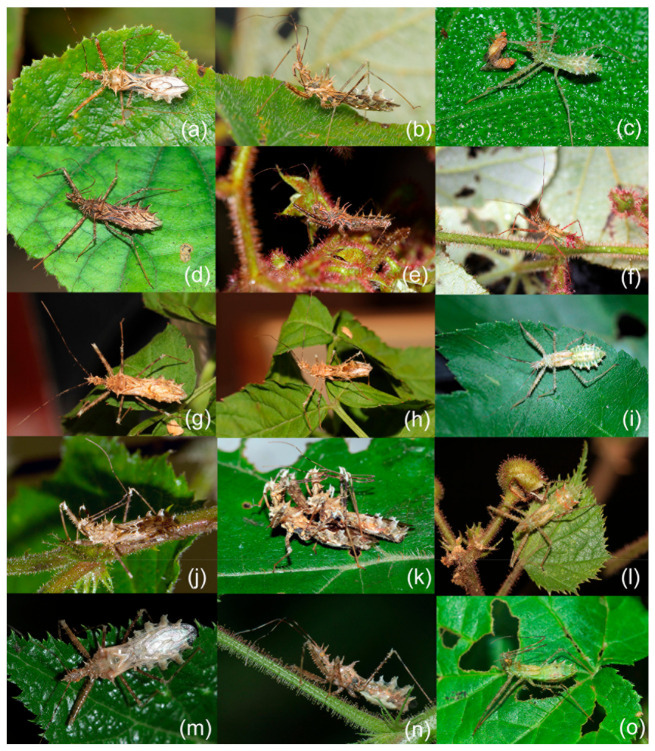
(**a**–**c**) *Sclomina erinacea* Stål, 1861, GZLS; (**d**–**f**) *Sclomina guangxiensis* Ren, 2001, GXLZ; (**g**–**i**) *Sclomina parva* sp. nov., SXYX; (**j**–**l**) *Sclomina xingrensis* sp. nov., GZXR; (**m**–**o**) *Sclomina pallens* sp. nov., GZTL. (**a**,**b**,**d**,**e**,**g**,**h**,**j**,**k**,**m**,**n**) adult; (**c**,**f**,**i**,**l**,**o**) nymph.

**Figure 7 insects-12-00251-f007:**
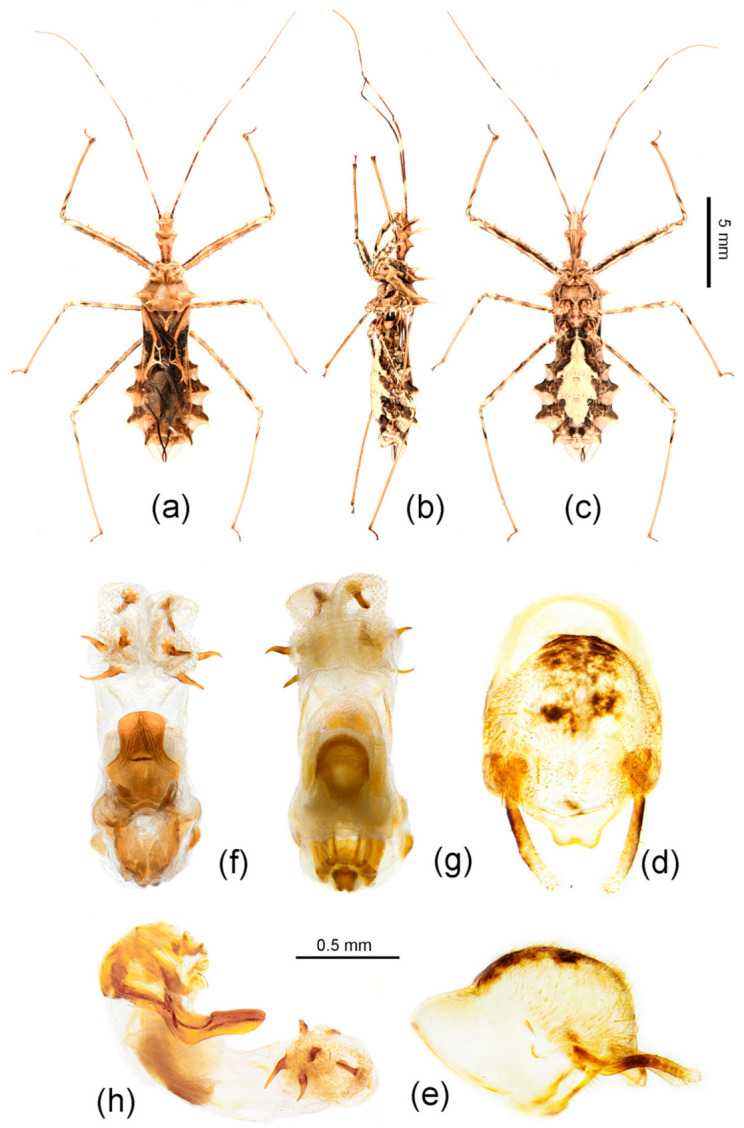
*Sclomina erinacea* Stål, 1861, (**a**–**c**) female, GZHP, habitus; (**d**–**h**) male, GZHP, male external genitalia. (**d**,**e**) Pygophore with paramere; (**f**–**h**) phallus. (**a**,**f**) dorsal view; (**b**,**e**,**h**) lateral view; (**c**,**d**,**g**) ventral view. Scale bar of (**a**–**c**) = 5 mm, (**d**–**h**) = 0.5 mm.

**Figure 8 insects-12-00251-f008:**
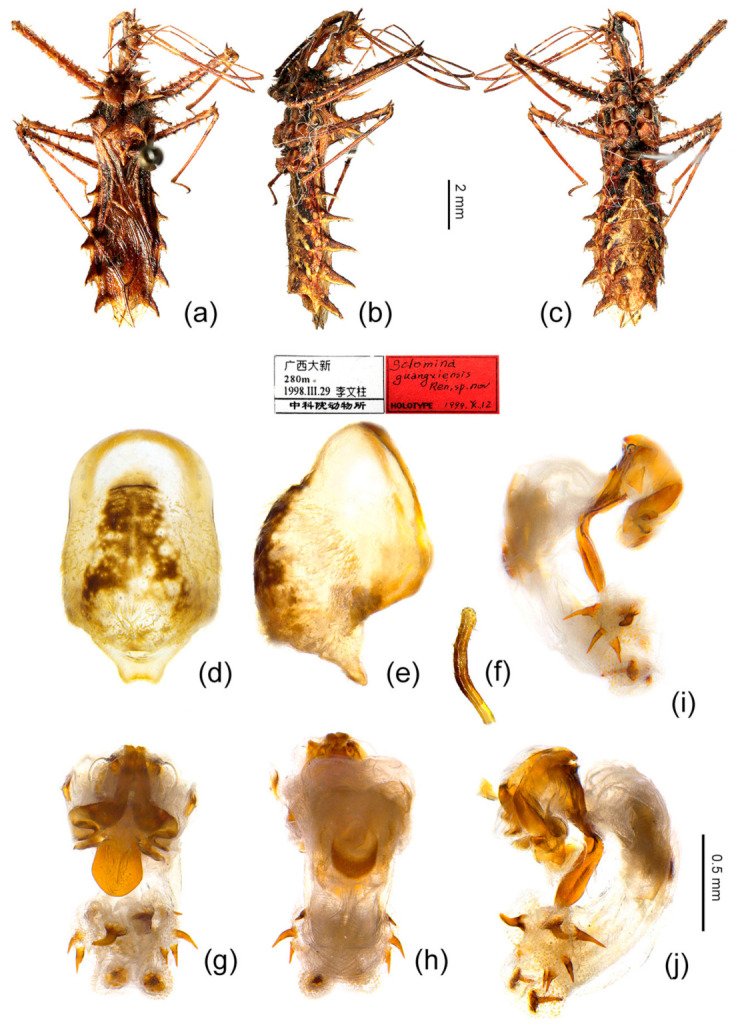
*Sclomina guangxiensis* Ren, 2001. (**a**–**c**) male, Holotype, habitus; (**d**–**j**) male, GXLZ, male external genitalia. (**d**,**e**) pygophore; (**f**) paramere; (**g**–**j**) phallus. (**a**,**g**) dorsal view; (**b**,**e**,**i**,**j**) lateral view; (**c**,**d**,**h**) ventral view. Scale bar of (**a**–**c**) = 2 mm, (**d**–**j**) = 0.5 mm.

**Figure 9 insects-12-00251-f009:**
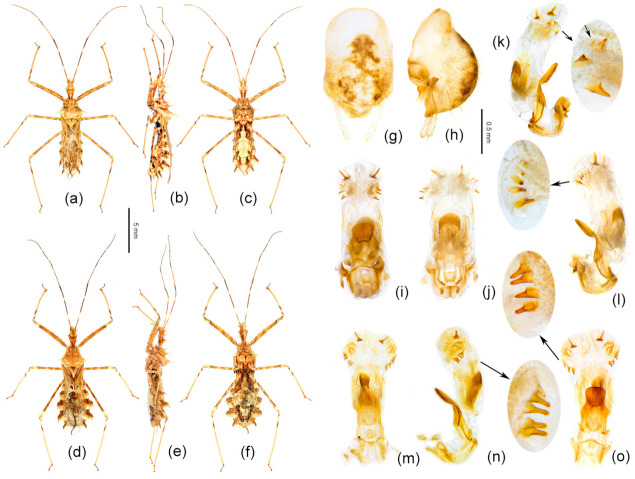
*Sclomina pallens* sp. nov., (**a**–**c**) male, (**d**–**f**) female, GZAL, habitus; (**g**–**l**) male, GXAL, male external genitalia; (**m**–**o**) male, GXTL, male external genitalia. (**g**,**h**) pygophore with paramere; (**i**–**o**) phallus. (**a**,**d**,**i**,**o**) dorsal view; (**b**,**e**,**h**,**k**,**l**,**n**) lateral view; (**c**,**f**,**g**,**j**,**m**) ventral view. Scale bar (**a**–**f**) = 5 mm, (**g**–**o**) = 0.5 mm. The subapical spines of (**k**,**l**,**n**,**o**) are enlarged.

**Figure 10 insects-12-00251-f010:**
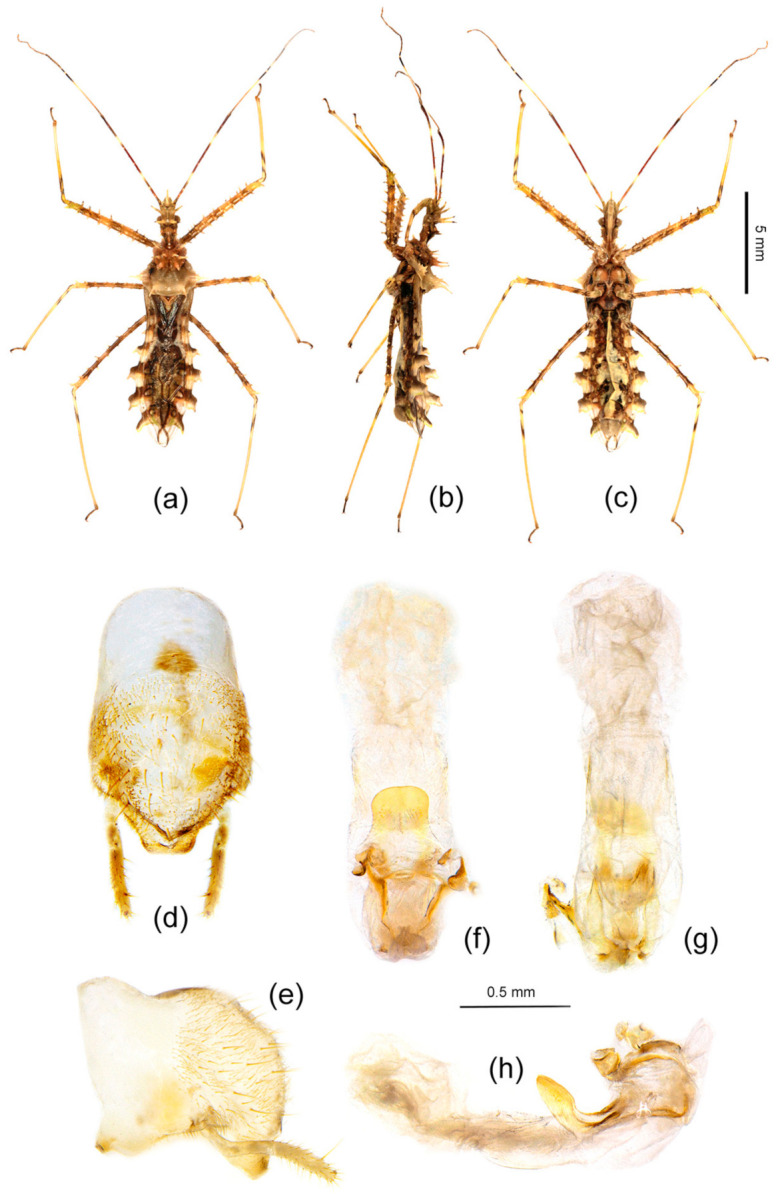
*Sclomina parva* sp. nov., (**a**–**c**) male, SXYX, habitus; (**d**–**h**) male external genitalia. (**d**,**e**) pygophore with paramere; (**f**–**h**) phallus. (**a**,**f**) dorsal view; (**b**,**e**,**h**) lateral view; (**c**,**d**,**g**) ventral view. Scale bar of (**a**–**c**) = 5 mm, (**d**–**h**) = 0.5 mm.

**Figure 11 insects-12-00251-f011:**
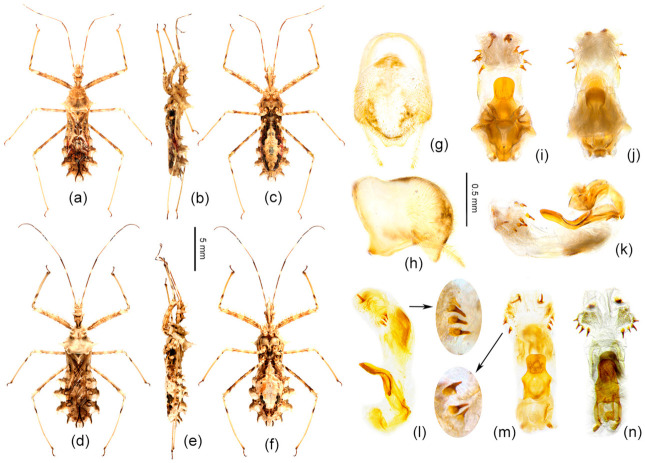
*Sclomina xingrensis* sp. nov., (**a**–**c**) male, (**d**–**f**) female, GZXR, habitus; (**g**–**k**) male, GZXR1, male external genitalia; (**l**–**m**) male, GZXR2, male external genitalia; (**n**) male, GZLS1, male external genitalia. (**g**,**h**) pygophore with paramere; (**i**–**n**) phallus. (**a**,**d**,**i**,**m**,**n**) dorsal view; (**b**,**e**,**h**,**k**,**l**) lateral view; (**c**,**f**,**g**,**j**) ventral view. Scale bar of (**a**–**f**) = 5 mm, (**g**–**n**) = 0.5 mm. The sub-apical spines of (**l**,**m**) are enlarged.

## Data Availability

All sequences were deposited in the GenBank under accession numbers of MW387643–MW387949.

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
