# Peer review of "Integrative Taxonomy of the Spinous Assassin Bug Genus Sclomina (Heteroptera: Reduviidae: Harpactorinae) Reveals Three Cryptic Species Based on DNA Barcoding and Morphological Evidence"

_insects, 2021, doi:10.3390/insects12030251_

Round 1
Reviewer 1 Report
The text of the MS needs many corrections; especially its taxonomic part should be fixed according to the Code (ICZN) requirements – see the comments placed directly to the text.
Redescriptions of known species and descriptions of new species should be in the passive voice. They should include "Diagnostic characters" (or "Diagnosis") and "Description" ("Description" and "Structure" should be combined into a single paragraph).
Moreover, when a new species is described, the author's name (with its surname's first letter) should be assigned to the new species when two authors with the same family name authored the paper (I mean P. Zhao and Q. Zhao).
Other comments were put directly to the text.

Reviewer 2 Report
This is a study on the lineage and classification of assassin bugs (genus Sclomina) in a wide area of China including remote islands, and in addition to the descriptive classification of morphological characteristics, the DNA barcoding is also conducted. Those results are in agreement with each other, and I'd like to evaluate highly as a comprehensive taxonomic research. From such point of views, I would like to recommend that this manuscript be accepted by the MDPI "Insects" journal.
However, some of the details need to be improved as listed below.
Generally, for the Insect DNA barcoding is used the sequence of the 658 base portion of the mitochondrial DNA COI region. In this study, the 852-bp sequence data including the the mtDNA COI region were obtained, but only 658-bp sequences were used for their phylogenetic analyses. Why did the authors shrink their dataset to a smaller one?
L232: Bootstrap support: Not the “Bayesian inference”, bootstrap probability of the NJ tree?
L246: distance P > p-distance?
Many scientific names should be italicized:
L405, L481×2, L440-445, L452, L488, L540, L546, L599, L656 (Highlighted with yellow markers)
Figure 1: What does the legend label (0.505-1) mean? Probably I think that the Bayesian posterior probability (BPP), but it needs explanation. In addition, looking at the color of the branches of the phylogenetic tree, the nodes with high BPP are blue, but the area around 1 in the legend is purple. In other words, I think there is a discrepancy between the figure and the legend.
Figure 2: Regarding the notation of the band part of the divergence periods shown below the phylogenetic tree, is it "0 Mya" instead of "0 My"? It is stated in the "Discussion" section that the populations of Hainan Island and Taiwan Island in Clade A (S. eronacea) are genetically differentiated, however I was not possible to read which OUTs of the phylogenetic tree corresponds to them. Therefore, the Figure 1 (c) needs to be more enlarged and displayed. Therefore, the Figure 1 (c) needs to be more magnified so that the OTU names can be read properly. In addition, I am very interested about the phylogenetic position within the Clade A of one population located of northern region of the Changjiang River. Both will be resolved by zooming in on the OTU name.

Round 2
Reviewer 1 Report
Almost all of my suggestions were accepted. However, the following should still be corrected:
Line 632: change “Paratype” to “Paratypes”
Line 714: change “adulst’s” to „adult’s”
In Sclomina xingrensis P. Zhao & Cai sp. nov., "Description" and "Structure" weren’t combined into a single paragraph, as was correctly done for all other species (please correct).
